# (D-Ala^2^)GIP Inhibits TNF-α-Induced Osteoclast Formation and Bone Resorption, and Orthodontic Tooth Movement

**DOI:** 10.3390/ijms27010199

**Published:** 2025-12-24

**Authors:** Angyi Lin, Hideki Kitaura, Jinghan Ma, Fumitoshi Ohori, Aseel Marahleh, Kayoko Kanou, Kohei Narita, Ziqiu Fan, Kou Murakami, Hiroyasu Kanetaka

**Affiliations:** 1Division of Orthodontics and Dentofacial Orthopedics, Tohoku University Graduate School of Dentistry, 4-1 Seiryo-Machi, Aoba-ku, Sendai 980-8575, Miyagi, Japan; lin.angyi.r5@dc.tohoku.ac.jp (A.L.); ma.jinghan.c1@tohoku.ac.jp (J.M.); fumitoshi.ohori.b4@tohoku.ac.jp (F.O.); kayoko.kano.b1@tohoku.ac.jp (K.K.); kohei.narita.a2@tohoku.ac.jp (K.N.); fan.ziqiu.q1@dc.tohoku.ac.jp (Z.F.); kou.murakami.b2@tohoku.ac.jp (K.M.); hiroyasu.kanetaka.e6@tohoku.ac.jp (H.K.); 2Creative Interdisciplinary Research Division, Frontier Research Institute for Interdisciplinary Sciences, Tohoku University, 6-3 Aramaki Aza Aoba, Aoba-ku, Sendai 980-8578, Miyagi, Japan; aseel.mahmoud.suleiman.marahleh.e6@tohoku.ac.jp; 3Division of Advanced Dental Science and Technology, Graduate School of Biomedical Engineering, Tohoku University, 6-6-12, Aramaki Aza Aoba, Aoba-ku, Sendai 980-8579, Miyagi, Japan; 4Liaison Center for Innovative Dentistry, Division of Interdisciplinary Co-Creation (ICC-Division), Tohoku University Graduate School of Dentistry, 4-1 Seiryo-Machi, Aoba-ku, Sendai 980-8575, Miyagi, Japan

**Keywords:** GIP, TNF-α, osteoclast, bone resorption, orthodontic tooth movement, root resorption

## Abstract

The incretin hormone glucose-dependent insulinotropic polypeptide (GIP) promotes insulin secretion, lowers blood glucose levels, and is increasingly linked to bone remodeling. Native GIP is quickly inactivated by the enzyme dipeptidyl peptidase-4 (DPP-4), whereas (D-Ala^2^)GIP is a novel GIP analog engineered to resist DPP-4 degradation. Tumor necrosis factor-alpha (TNF-α), a key proinflammatory cytokine, promotes osteoclastogenesis and is notably upregulated during orthodontic tooth movement (OTM). This study aimed to evaluate the effects of (D-Ala^2^)GIP on TNF-α-induced osteoclast formation and bone resorption in vivo, as well as on OTM and related root resorption. Mice received daily supracalvarial injections of TNF-α with or without (D-Ala^2^)GIP for 5 days. The (D-Ala^2^)GIP-treated group showed significantly reduced osteoclast formation, bone resorption, and expression of osteoclastic markers TRAP and cathepsin K, compared to the group that received TNF-α alone. OTM was induced in mice by applying a nickel-titanium closed-coil spring, and mice were treated with either phosphate-buffered saline (PBS) or (D-Ala^2^)GIP every 2 days. After 12 days, the (D-Ala^2^)GIP-treated group showed significantly reduced tooth movement and fewer osteoclasts and odontoclasts on the compression side compared to the PBS control. These findings suggest that (D-Ala^2^)GIP inhibits OTM, potentially by suppressing TNF-α-driven osteoclastogenesis and bone resorption.

## 1. Introduction

Diabetes mellitus is a growing global health concern, characterized by chronic hyperglycemia due to absolute or relative deficiencies in insulin production or action [1,2,3]. The number of patients with diabetes is increasing worldwide and is projected to exceed 1.31 billion by 2050, highlighting the need for more effective medications [4]. Glucose-dependent insulinotropic polypeptide (GIP), along with glucagon-like peptide-1 (GLP-1), is one of the principal incretin hormones that enhances insulin secretion in response to nutrient intake [5,6]. Secreted by intestinal K cells, GIP acts by binding to its receptor (GIPR) on pancreatic β-cells, thereby promoting insulin release and reducing blood glucose levels [7,8]. However, native GIP is rapidly degraded in the circulation by dipeptidyl peptidase-4 (DPP-4) [9,10]. Several GIP analogs and receptor agonists with enhanced resistance to DPP-4 have been developed, including (D-Ala^2^)GIP—a modified form of GIP with an alanine at position 2 [11,12].

Bone tissue undergoes continuous remodeling, a dynamic process involving the coordinated activities of osteoclasts, which resorb bone, and osteoblasts, which form new bone [13,14,15]. GIP regulates bone remodeling as part of the gut-bone axis [16,17]. GIPR is expressed in osteoclasts, osteoblasts, and osteocytes [18,19]. In vitro, GIP treatment reduces bone-resorptive activity in human osteoclasts through activation of GIPR [20]. GIP dose-dependently inhibits pit formation primed by receptor activator of nuclear factor kappa-B ligand (RANKL) in mouse osteoclast-like cells [18]. In vivo, intravenous infusion of GIP suppresses serum C-terminal telopeptide of type I collagen (CTX-1) levels, a biochemical marker reflecting the rate of osteoclastic bone resorption in patients with diabetes and their healthy counterparts [21,22,23]. Intraperitoneal administration of N-AcGIP, another long-acting DPP-4-resistant GIP analog, alleviated tibial bone degradation in ovariectomized mice [24]. We previously reported that subcutaneous injection of (D-Ala^2^)GIP, the GIP analog used in the present study, inhibited lipopolysaccharide (LPS)-triggered acute inflammatory osteoclast formation and bone resorption in mouse calvariae [25].

Tumor necrosis factor-α (TNF-α) is a cytokine that exerts pleiotropic effects on various cell types via two cell-surface receptors: TNF receptor type 1 (TNFR1) and TNF receptor type 2 (TNFR2) [26,27]. TNF-α induces osteoclastogenesis directly or by upregulating RANKL expression in stromal cells, osteoblasts, and osteocytes [28,29,30]. Osteoclast formation mediated by TNF-α contributes to osteoporosis in inflammatory bone disorders, including rheumatoid arthritis and periodontal disease [31,32]. Orthodontic tooth movement (OTM) is achieved by the remodeling of the periodontal ligament and alveolar bone under mechanical loading, with osteoclastic bone resorption occurring on the compression side and osteoblastic bone formation taking place on the tension side [33,34,35]. Root resorption, primarily mediated by odontoclasts during orthodontic tooth movement, is a localized inflammatory process characterized by the degradation of mineralized dental tissue—including cementum and dentin—in response to mechanical stress, and is often considered an unavoidable side effect of orthodontic treatment [36,37,38,39]. Elevated TNF-α concentrations were detected in human gingival crevicular fluid 24 h after OTM [40]. Similarly, in mice, TNF-α expression was confirmed by immunohistochemistry on days 2, 6, and 10 of OTM on the compression side of periodontal tissue [41]. Furthermore, TNF-α induces RANKL expression in osteocytes during OTM [42]. In TNF receptor-deficient mice, both the distance of OTM and osteoclastogenesis on the compression side were significantly suppressed [43]. In aged mice, OTM was reduced due to a decrease in TNF-α-induced osteoclast formation [44]. These results suggest the pivotal role of TNF-α in promoting OTM.

We previously reported that (D-Ala^2^)GIP inhibited TNF-α-induced osteoclastogenesis in vitro [25]. However, the effect of GIP signaling on TNF-α-induced osteoclast formation and bone resorption in vivo remains unknown. Research on the effect of GIP on orthodontic tooth movement remains limited. A previous study demonstrated that deletion of GIP signaling enhanced OTM. In the GIPR knockout (GIPRKO) mouse model, both the distance of OTM and osteoclast formation on the compression side were significantly increased, suggesting that endogenous GIP signaling inhibits tooth movement [45]. However, the effect of exogenous GIP application on this force-loading-induced bone remodeling model has not yet been reported. Therefore, in this study, we used the DPP-4-resistant GIP analog (D-Ala^2^)GIP to investigate the effects of exogenous and prolonged GIP signaling on TNF-α-induced osteoclastogenesis and bone resorption in vivo, as well as on orthodontic tooth movement and associated root resorption.

## 2. Results

### 2.1. (D-Ala^2^)GIP Inhibited TNF-α-Induced Osteoclastogenesis In Vivo

We adopted a TNF-α injection model targeting mouse calvariae to assess whether (D-Ala^2^)GIP could suppress TNF-α-induced osteoclastogenesis in vivo. Histological analysis with tartrate-resistant acid phosphatase (TRAP) staining revealed a robust induction of TRAP-positive multinucleated osteoclasts along the central mesenchymal suture in the TNF–α–treated group compared to phosphate-buffered saline (PBS) or (D-Ala^2^)GIP controls. However, co-administration of TNF-α with (D-Ala^2^)GIP significantly reduced osteoclast numbers (Figure 1a,b). Consistent with these histological findings, real-time RT-PCR analysis showed that mRNA expression levels of osteoclastic markers TRAP and cathepsin K were significantly elevated in the TNF-α group but were markedly diminished in the group receiving both TNF-α and (D-Ala^2^)GIP, suggesting that (D-Ala^2^)GIP effectively antagonized TNF-α-driven osteoclast formation (Figure 1c,d).

### 2.2. (D-Ala^2^)GIP Inhibited TNF-α-Induced Bone Destruction In Vivo

The extent of TNF-α-induced inflammatory bone destruction was evaluated by performing micro-computed tomography (micro-CT) imaging on mouse calvariae. The TNF-α-treated group exhibited the greatest area of bone resorption among all four groups. However, mice that received both TNF-α and (D-Ala^2^)GIP injections exhibited a significant reduction in bone resorption compared to the TNF-α-only group, indicating that (D-Ala^2^)GIP notably dampened bone destruction induced by TNF-α (Figure 2a,b).

### 2.3. (D-Ala^2^)GIP Dampened Orthodontic Tooth Movement

The effect of (D-Ala^2^)GIP on orthodontic tooth movement was evaluated by subjecting mice to mechanical loading for 12 days while administering either PBS or (D-Ala^2^)GIP injections (Figure 3a). After the 12-day force application, the mesial movement of the maxillary left first molar was measured as the distance between the first and second molars in silicone impressions (Figure 3b). In the PBS-treated group, the average tooth movement distance was 147.31 ± 26.07 μm, whereas in the (D-Ala^2^)GIP-treated group, the distance was reduced to 75.25 ± 8.55 μm, suggesting that local administration of (D-Ala^2^)GIP significantly suppressed the distance of orthodontic tooth movement (Figure 3c).

### 2.4. (D-Ala^2^)GIP Dampened Osteoclast Formation During Tooth Movement

The cellular mechanisms underlying the reduction in tooth movement were explored by assessing osteoclast formation using TRAP staining. Tissue sections were prepared at five levels beneath the root bifurcation of the maxillary left first molar distobuccal root (M1DB) in both groups. The compression side was the focus of histological evaluation (Figure 4a). In the PBS group, 12-day force loading led to a marked increase in the number of TRAP-positive multinucleated cells. In contrast, mice treated with (D-Ala^2^)GIP exhibited significantly fewer TRAP-positive cells in the same region, indicating that (D-Ala^2^)GIP inhibited osteoclast formation on the compression side during tooth movement (Figure 4b).

### 2.5. (D-Ala^2^)GIP Dampened Orthodontic Tooth Movement-Induced Odontoclast Formation and Root Resorption

Similarly, TRAP staining was employed to assess the impact of (D-Ala^2^)GIP on root resorption associated with orthodontic tooth movement. Odontoclasts located on the root surface of the mesial side of the distobuccal root of the first molar (M1DB) were identified and quantified (Figure 5a). In the PBS-treated group, severe root resorption was observed, as indicated by a high number of TRAP-positive odontoclasts along the root surface. In contrast, mice treated with (D-Ala^2^)GIP exhibited a marked reduction in the number of odontoclasts (Figure 5b). We also evaluated the percentage of root resorption surface induced by OTM in the two groups. Histology sections revealed a large area of resorption along the root surface in the PBS group, whereas the (D-Ala^2^)GIP-treated group showed a significant reduction in resorbed surface area (Figure 5c,d). The results suggest that (D-Ala^2^)GIP effectively suppressed OTM-induced root resorption by inhibiting odontoclast formation on the compression side.

## 3. Discussion

In this study, we first evaluated the effect of (D-Ala^2^)GIP on TNF-α-induced osteoclastogenesis in vivo. Mice that received TNF-α with (D-Ala^2^)GIP demonstrated significantly fewer osteoclasts and reduced bone destruction compared with mice administered only TNF-α after 5 consecutive days of supracalvarial injection, suggesting that (D-Ala^2^)GIP dampens TNF-α-primed osteoclast formation and bone resorption. We next explored the impact of (D-Ala^2^)GIP on orthodontic tooth movement. The results revealed that the application of (D-Ala^2^)GIP suppressed tooth movement distance after the 12-day force-loading period. The suppressive effect of (D-Ala^2^)GIP on OTM might be mediated by inhibiting osteoclastogenesis on the compression side, induced by TNF-α produced during OTM.

Diabetes mellitus, a major lifestyle-related disease, has seen a rapid rise in prevalence worldwide. Patients with diabetes demonstrate significantly higher blood glucose levels and impaired insulin secretion or action [1,2,3]. The incretin hormone GIP promotes insulin secretion after binding to its receptor, GIPR, on pancreatic β cells, thereby decreasing blood glucose levels [5,6,7,8]. In bone, GIPR is expressed in osteoblasts, osteoclasts, and osteocytes [18,19], and GIP signaling is increasingly recognized as a key modulator of bone metabolism as part of the enteroendocrine-osseous axis [16,17]. Endogenous GIP contributes significantly to the postprandial suppression of bone resorption in humans [46]. Exogenous GIP infusion has also been shown to reduce serum CTX levels in both healthy individuals and patients with diabetes [21,22,23]. In our previous study, we demonstrated that (D-Ala^2^)GIP, a long-acting GIP analog, significantly ameliorated LPS-induced short-term inflammatory bone resorption in mouse calvariae [25].

TNF-α is a pro-inflammatory cytokine that regulates bone remodeling. It promotes osteoclastogenesis by upregulating RANKL synthesis in marrow stromal cells, osteoblasts, and osteocytes [29,30], or by directly stimulating osteoclast formation independent of RANKL [28]. Inflammatory bone disorders, such as rheumatoid arthritis and periodontitis, are often characterized by TNF-α-driven excessive osteoclast formation, which contributes to pathological bone loss [31,32]. We previously reported that (D-Ala^2^)GIP directly reduces TNF-α-induced osteoclast differentiation in murine bone marrow-derived macrophages in vitro [25]. In this study, we aimed to investigate whether (D-Ala^2^)GIP also inhibits osteoclastogenesis primed by TNF-α in vivo, using a mouse calvaria model for evaluation. We conducted supracalvarial injections of TNF-α with or without (D-Ala^2^)GIP for 5 consecutive days. Based on our earlier study using 25 nmol/kg b.w. of (D-Ala^2^)GIP to inhibit LPS-triggered bone resorption, we applied the same dosage here [25]. Histology sections showed that osteoclast numbers were significantly reduced in mice treated with both TNF-α and (D-Ala^2^)GIP, compared to those that received TNF-α only. The expression levels of essential osteoclast markers, TRAP and cathepsin K, were also reduced in mice treated with both TNF-α and (D-Ala^2^)GIP compared to those exposed to TNF-α alone. In addition, we assessed bone destruction by micro-CT by calculating the resorption area within a defined region of interest (ROI) to quantify local bone resorption in the calvaria. The group treated with both TNF-α and (D-Ala^2^)GIP showed less bone resorption than the TNF-α-only group. The results suggest that (D-Ala^2^)GIP inhibits TNF-α-induced osteoclast formation and bone destruction in vivo, which is consistent with previous studies.

OTM occurs through alveolar bone remodeling, with bone resorption by osteoclasts on the compression side and bone formation by osteoblasts on the tension side [33,34,35]. Earlier work from our team highlighted the essential role of TNF-α in OTM. TNF-α expression was detected by immunohistochemistry in the periodontium on the compression side of the M1DB on days 2, 6, and 10 [41]. RANKL expression is induced by TNF-α in osteocytes during OTM [42]. Both osteoclast numbers and tooth movement were significantly reduced in TNF receptor-deficient mice, and a decrease in TNF-α-mediated osteoclast formation resulted in diminished OTM in aged mice [43,44]. These results suggest that orthodontic force promotes TNF-α expression, which in turn facilitates osteoclastogenesis and enhances bone resorption in periodontal tissue, thereby accelerating tooth movement. We thereby hypothesized that, as (D-Ala^2^)GIP suppresses osteoclastogenesis and bone resorption induced by TNF-α in our in vivo calvaria model, it might also exert an inhibitory effect on orthodontic tooth movement.

In this study, we used a Ni-Ti closed-coil spring to induce mesial movement of the maxillary left first molar in mice. We aimed to assess the effect of (D-Ala^2^)GIP on OTM by injecting either PBS (as a control) or (D-Ala^2^)GIP every 2 days into the gingivobuccal fold of the first molar. We also applied the same dose of (D-Ala^2^)GIP, 25 nmol/kg b.w., as used in previous experiments. After the 12-day force loading, the average OTM distance was 147.31 ± 26.07 μm in the PBS group, which is consistent with our previous work and validates the reliability of the OTM model we used [47]. However, the distance decreased to 75.25 ± 8.55 μm in the (D-Ala^2^)GIP-administered group, which is approximately half of that in the PBS group, indicating a remarkable suppression of OTM by (D-Ala^2^)GIP. TRAP staining of histology sections demonstrated that in the (D-Ala^2^)GIP group, the number of osteoclasts on the compression side of the upper-left first molar distobuccal root is also downregulated compared with the PBS group. A previous study involving GIPRKO mice also implicated GIP signaling as a negative regulator of OTM. In contrast to wild-type mice, GIPRKO mice exhibited significantly increased tooth movement and a higher number of osteoclasts on the compression side [45]. Therefore, our findings align with observations in the GIPRKO model and further clarify that the exogenous activation of GIP signaling inhibits both orthodontic tooth movement and osteoclastogenesis induced by mechanical force. We thus propose that the inhibitory effect of (D-Ala^2^)GIP on OTM may be attributed to its suppression of osteoclast formation and bone resorption mediated by TNF-α, which is generated upon orthodontic force. Nevertheless, the underlying cellular mechanisms of this inhibitory effect remain to be fully elucidated. At present, direct evidence for GIPR expression on osteoclast precursors is lacking. And we did not investigate the expression of GIPR in osteoclast precursors in periodontal ligament during OTM in the present study. However, our previous in vitro study showed that (D-Ala^2^)GIP directly suppressed TNF-α-induced osteoclast formation in osteoclast precursor monocultures [25], supporting the possibility that GIP signaling can act directly on osteoclast precursors to inhibit osteoclastogenesis. In addition to reducing osteoclast number, GIP signaling can also attenuate osteoclast activity [18]. In the present study, the expression of osteoclast markers such as TRAP and cathepsin K, both closely associated with osteoclast resorptive function, was significantly downregulated, suggesting a functional suppression of osteoclast-mediated resorption. Therefore, the combined inhibition of osteoclast formation and activity likely contributed to the reduced TNF-α-mediated resorption observed in vivo. Moreover, indirect mechanisms may also have participated. We previously reported that (D-Ala^2^)GIP suppressed LPS-induced TNF-α expression in macrophages and reduced RANKL expression in osteoblasts in vitro [25]. These findings raise the possibility that immune cells and osteoblast lineage cells may also participate in mediating the effects of GIP signaling under inflammatory or mechanically loaded conditions. However, because the OTM and calvaria models employed in the present study differ from prior in vitro LPS-based systems, further studies are required to confirm contributions of these indirect pathways and clarify the cell-type-specific actions of GIP signaling during OTM and TNF-α-induced bone resorption.

Root resorption inevitably occurs during orthodontic treatment. Odontoclast-mediated root resorption has been reported to share a common cellular pathway with osteoclast-mediated bone resorption [36,37,38,39]. In this study, marked root resorption was observed in the PBS group after 12 days of force loading, as evidenced by numerous odontoclasts along the mesial side of the distobuccal root and a high percentage of root resorption surface. In contrast, the (D-Ala^2^)GIP group exhibited significantly fewer odontoclasts and a reduced resorptive surface ratio, suggesting that (D-Ala^2^)GIP suppresses odontoclast formation and thereby attenuates root resorption induced by orthodontic tooth movement.

In this study, we revealed the inhibitory effect of (D-Ala^2^)GIP on TNF-α-induced osteoclast formation and bone resorption in vivo, highlighting the potential of targeting GIP signaling as a therapeutic strategy for TNF-α-associated inflammatory bone diseases. Notably, this is the first report to demonstrate that exogenous and prolonged activation of GIP signaling can inhibit orthodontic tooth movement. Our findings also suggest that the underlying mechanism of this suppression is that (D-Ala^2^)GIP may reduce OTM by disrupting TNF-α-driven osteoclastogenesis and bone resorption, processes known to be upregulated in response to orthodontic force. With the increasing global prevalence of diabetes, the number of orthodontic patients with diabetes is also rising [48]. Tirzepatide, a widely used medication for treating type 2 diabetes and weight loss, functions as a dual incretin receptor agonist, activating both GLP-1 and GIP receptors [49]. Our data demonstrated that exogenous activation of GIP signaling attenuated tooth movement and markedly suppressed odontoclast-mediated root resorption, suggesting a potential protective effect of GIP on dental tissues during OTM. However, the present study employed a local injection model, which does not recapitulate the pharmacokinetics, systemic distribution, or receptor activation profiles produced by systemic medications such as tirzepatide. Therefore, our findings cannot be directly extrapolated to systemic drug effects in humans. Nonetheless, the biological actions observed here provide a foundation for future research on whether such systemic GIP-related therapies influence orthodontic biomechanics or root resorption in clinical contexts. In addition, this study was conducted in a healthy animal model. Future research should investigate how GIP signaling influences orthodontic responses under systemic conditions such as diabetes or obesity, which are major indications for incretin therapy. Overall, our findings offer novel insights into the role of the incretin hormone GIP in modulating biological responses during TNF-α-induced inflammatory bone conditions, as well as in the context of orthodontic tooth movement. A major strength of the study is that it is the first to directly evaluate the effects of exogenous GIP administration on orthodontic tooth movement and root resorption, thereby providing novel insight into the potential relevance of GIP signaling in orthodontic biology, while also demonstrating its modulatory role in TNF-α-associated inflammatory bone resorption in vivo. In contrast, the study also has several limitations, including the lack of mechanistic investigation into how GIP inhibits TNF-α-induced osteoclastogenesis, as well as the use of a local injection model rather than systemic drug administration. Future studies addressing these limitations will be essential to clarify the underlying mechanisms and to better define the biological role of GIP signaling in orthodontic and inflammatory bone contexts.

## 4. Materials and Methods

### 4.1. Animals and Reagents

Male C57BL/6J mice were purchased from CLEA Japan (Tokyo, Japan) and housed at the animal facility of Tohoku University. For the supracalvarial injection experiment, 8-week-old mice were used and maintained on a standard laboratory chow diet. For the orthodontic tooth movement experiment, 8-10-week-old mice were used and fed a granular diet (CLEA Japan) after appliance placement. Water was provided ad libitum in both cases. All animal care and experimental procedures were approved by and conducted in accordance with the Regulations for Animal Experiments and Related Activities at Tohoku University under the supervision of the Institutional Animal Care and Use Committee of the Tohoku University Environmental & Safety Committee. (D-Ala^2^)GIP was obtained from Bachem (Bubendorf, Switzerland). Recombinant murine TNF-α was prepared as previously described [47].

### 4.2. Histological Evaluation

Mice were randomly assigned to four groups (n = 4 per group) and received supracalvarial injections of PBS, TNF-α (3 μg/day), TNF-α (3 μg/day) + (D-Ala^2^)GIP (25 nmol/kg b.w.), or (D-Ala^2^)GIP (25 nmol/kg b.w.) for five consecutive days. On the sixth day, mice were sacrificed by cervical dislocation, and the calvariae were harvested. Separately, another cohort of mice (n = 4 per group) was subjected to orthodontic tooth movement treatment. After completion of OTM, maxillae from these mice were collected for histological analysis. Both in vivo experiments were performed once, and individual animals were treated as biological replicates.

Following resection, both the calvariae and maxillae were fixed in 4% PBS-buffered formaldehyde for 3 days at 4 °C. The calvariae were decalcified in 14% ethylenediaminetetraacetic acid (EDTA) for 30 days. After decalcification, samples were dehydrated using a tissue processor (TP1020, Leica, Wetzlar, Germany), embedded in paraffin, and sectioned coronally at a thickness of 5 μm using a microtome (REM-710·SB, Yamato Kohki Industrial Co., Ltd., Saitama, Japan). For orthodontic tooth movement analysis, the maxillae were decalcified in 14% EDTA for 30 days, followed by dehydration with the same tissue processor. Then, the paraffin-embedded maxilla tissues were sectioned transversely at a thickness of 4 μm.

The formation of osteoclasts and odontoclasts was evaluated by subjecting both calvaria and maxilla sections to TRAP staining. The TRAP solution was prepared using sodium acetate buffer (pH 5.0), naphthol AS-MX phosphate (Sigma-Aldrich, St. Louis, MO, USA), Fast Red Violet LB Salt (Sigma-Aldrich, St. Louis, MO, USA), and sodium tartrate. After staining, sections were counterstained with hematoxylin. Osteoclasts were identified as TRAP-positive multinucleated cells (≥3 nuclei) located along bone surfaces. For the calvariae, osteoclasts were counted along the central mesenchymal suture. For the maxillae, osteoclasts were assessed along the alveolar bone of the mesial (compression) side of M1DB, a site known for active osteoclastogenesis and odontoclastogenesis during tooth movement, at five levels beneath the root bifurcation (100, 140, 180, 220, and 260 μm). Similarly, odontoclasts were counted as TRAP-positive multinucleated cells (≥3 nuclei) located along the root surface of M1DB. The root resorption surface was identified as areas of cementum discontinuity, thinning, or resorption lacunae in contact with odontoclasts. In transverse sections of M1DB, the mesial (compression) surface of the root was traced manually along the cementum-periodontal ligament interface using ImageJ software (version 1.53, NIH, Bethesda, MD, USA) to obtain the total mesial surface length (solid line in Figure 5c). Within the same contour, the segments corresponding to resorption lacunae were traced separately as the resorption surface (interrupted line in Figure 5c). The histologic resorption surface percentage was calculated as the resorption surface length divided by the total mesial side root surface length.

### 4.3. Micro-CT Investigation of Bone Resorption Area

Bone resorption in the calvariae was evaluated by performing micro-CT scanning using a high-resolution scanner (ScanXmate-E090; Comscantecno Co., Ltd., Kanagawa, Japan) after the samples were fixed in 4% PBS-buffered formaldehyde. The acquired raw data were reconstructed into three-dimensional images using TRI/3DBON64 0.9.0.0 software (RATOC System Engineering, Tokyo, Japan). Bone resorption was identified as radiolucent (black) areas within the calvarial bone and was particularly assessed in a 50 × 70-pixel rectangular ROI centered at the intersection of the sagittal and coronal sutures. The extent of bone resorption was quantified by calculating the proportion of the resorbed area relative to the total calvaria surface area within the ROI, using ImageJ software (NIH, Bethesda, MD, USA).

### 4.4. RNA Isolation and Real-Time RT-PCR Analysis

Bone chips from calvariae were collected immediately after sacrifice and frozen in liquid nitrogen. Total RNA was extracted by homogenizing the samples in 800 μL TRIzol reagent (Invitrogen, Carlsbad, CA, USA) using Micro Smash MS-100R (Tomy Seiko, Tokyo, Japan), followed by purification with the RNeasy Mini Kit (Qiagen, Valencia, CA, USA) according to the manufacturer’s protocol. Sample cDNA was synthesized from the isolated RNA using Superscript IV First-Strand Synthesis System (Invitrogen, Carlsbad, CA, USA). The expression levels of TRAP and cathepsin K were analyzed by real-time RT-PCR using Bio-Rad CFX96 Real-Time System in conjunction with the C1000 Thermal Cycler (Bio-Rad, Hercules, CA, USA). The reaction system consisted of 2 μL of cDNA sample, 23 μL SYBR Premix Ex Taq (Takara Bio Inc., Shiga, Japan), and 50 pmol/μL primers. Sequences of primers were as follows: GAPDH: 5′-GGTGGAGCCAAAAGGGTCA-3′ and 5′-GGGGGCTAAGCAGTTGGT-3′; TRAP: 5′-AACTTGCGACCATTGTTA-3′ and 5′-GGGGACCTTTCGTTGATGT-3′; Cathepsin K: 5′-GCAGAGGTGTGTACTATGA-3′ and 5′-GCAGGCGTTGTTCTTATT-3′. GAPDH was used as the reference gene to normalize TRAP and cathepsin K expression.

### 4.5. Orthodontic Tooth Movement

Orthodontic tooth movement was induced as previously described [50]. Briefly, mice were randomly assigned to receive either PBS or (D-Ala^2^)GIP (n = 4 mice per group). Anesthesia was administered via intraperitoneal injection of a mixture of medetomidine, midazolam, and butorphanol. The orthodontic appliance was then installed in the oral cavity. A hole was drilled in the maxillary alveolar bone near the incisors on the palatal side using a 0.8 mm round steel bur near the incisors on their palatal side. A nickel-titanium closed coil spring (TOMY SEIKO Co., Ltd., Tokyo, Japan) was then fixed between the maxillary left first molar and the incisors using a 0.1 mm stainless steel wire through the hole. According to the manufacturer’s instructions, the appliance exerted a constant force of 10 g throughout the experiment. PBS or (D-Ala^2^)GIP (25 nmol/kg b.w.) was injected into the gingivobuccal fold of the upper-left first molar every 2 days (days 0, 2, 4, 6, 8, and 10) under anesthesia. Force-loading was maintained for a total of 12 days to induce orthodontic tooth movement.

### 4.6. Measurement of Tooth Movement

On day 12 following force application, mice were sacrificed, and the maxillae were carefully dissected. Impressions of the maxillary dentition were obtained using trays filled with hydrophilic vinyl polysiloxane material (EXAMIXFINE Injection Type, GC Co., Tokyo, Japan). The impressions were then used to measure the distance between the mesial surface of the second molar and the distal surface of the first molar, defined as the extent of orthodontic tooth movement. Measurements were performed under a stereo microscope (Leica M165 FC, Wetzlar, Germany) to ensure high-resolution and accurate evaluation.

### 4.7. Statistical Analysis

All data are presented as mean ± standard deviation (SD). Student’s *t*-test was used to compare differences between two groups, and for multiple comparisons, one-way analysis of variance (ANOVA) followed by the Tukey–Kramer test was performed. Statistical significance was set at *p* < 0.05.

## 5. Conclusions

In conclusion, the present study demonstrates that (D-Ala^2^)GIP suppresses TNF-α-induced osteoclast formation and bone resorption in vivo. It also inhibits orthodontic tooth movement, likely through the attenuation of osteoclastogenesis and bone destruction mediated by TNF-α on the compression side. In addition, (D-Ala^2^)GIP alleviated OTM-induced root resorption, suggesting a potential protective effect on dental tissues. These findings provide novel biological insight into the role of GIP signaling in both inflammatory bone conditions and orthodontic tooth movement. Although the present results cannot be directly extrapolated to systemic incretin-based therapies, they establish a foundation for future mechanistic and translational studies aimed at clarifying how GIP signaling influences bone and dental tissue responses under inflammatory or orthodontic loading conditions.

## Figures and Tables

**Figure 1 ijms-27-00199-f001:**
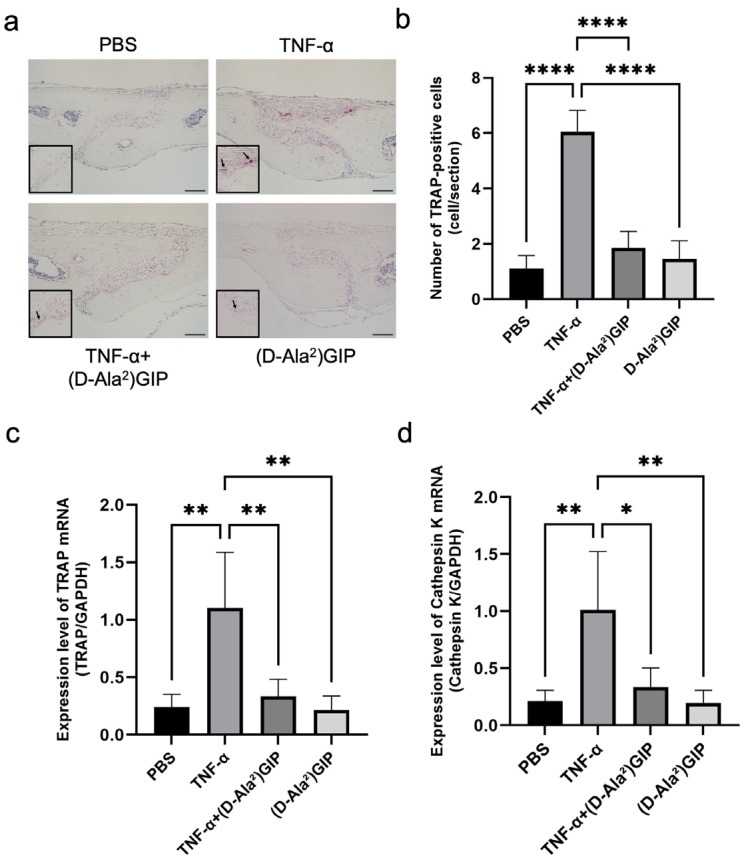
(D-Ala^2^)GIP suppressed TNF-α-induced osteoclastogenesis in vivo. (**a**) TRAP staining of mouse calvaria sections following subcutaneous injection of PBS, TNF-α, TNF-α+ (D-Ala^2^)GIP, or (D-Ala^2^)GIP alone for 5 days. Sections were counterstained with hematoxylin. Arrows indicate osteoclasts. Scale bar = 50 μm. (**b**) Quantification of TRAP-positive multinucleated cells in the mesenchymal tissue of the calvarial sagittal suture region. (**c**) Relative expression of TRAP mRNA in calvaria bone chips, normalized to GAPDH. (**d**) Relative expression of Cathepsin K mRNA in calvaria bone chips, normalized to GAPDH. Data are shown as mean ± SD. Statistical significance was determined using one-way ANOVA with Tukey–Kramer test (*n* = 4 mice/group; * *p* < 0.05, ** *p* < 0.01, **** *p* < 0.0001).

**Figure 2 ijms-27-00199-f002:**
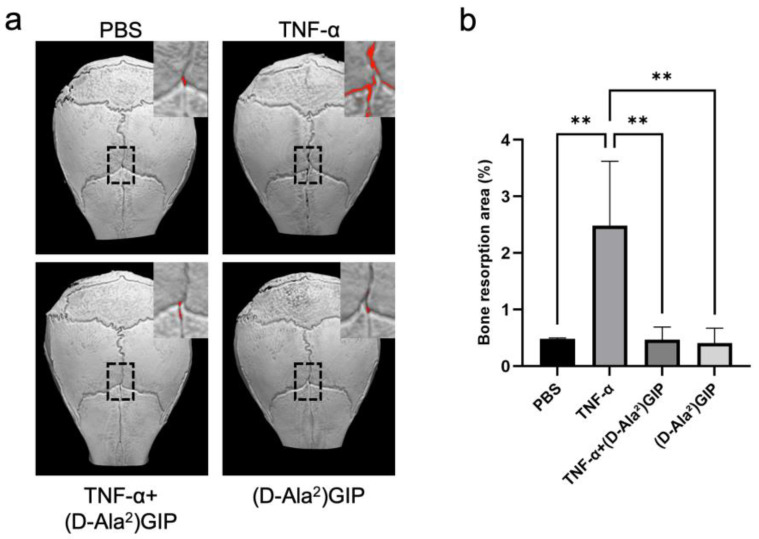
(D-Ala^2^)GIP suppressed TNF-α-induced bone resorption in vivo. (**a**) Representative 3D reconstructed images of mouse calvariae obtained by micro-CT after subcutaneous injection of PBS, TNF-α, TNF-α + (D-Ala^2^)GIP, or (D-Ala^2^)GIP alone for 5 days. Red areas indicate bone resorption sites. (**b**) The quantification of bone resorption is expressed as the percentage of the resorbed area relative to the 50x70 pixel ROI. Data are shown as mean ± SD. Statistical significance was determined using one-way ANOVA with Tukey–Kramer test (*n* = 4 mice/group; ** *p* < 0.01).

**Figure 3 ijms-27-00199-f003:**
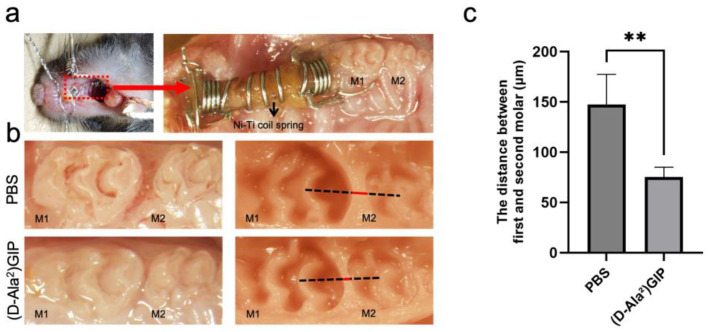
(D-Ala^2^)GIP inhibited orthodontic tooth movement after the 12-day experiment period. (**a**) Intraoral photo of the orthodontic appliance used. The appliance was fixed between the maxillary incisors and the left first molar. The red dashed box indicates the region of appliance placement, which is shown at higher magnification in the right panel. (**b**) Intraoral and silicone impression images after the 12-day tooth movement. The impressions were taken to measure the distance between the maxillary first (M1) and second molars (M2), which indicates tooth movement distance. The black dashed line represents the reference line used for distance measurement, while the red line indicates the measured distance between M1 and M2. (**c**) Distance of orthodontic tooth movement in PBS and (D-Ala^2^)GIP group. Data are shown as mean ± SD. Statistical significance was determined using unpaired *t*-test (*n* = 4 mice/group; ** *p* < 0.01).

**Figure 4 ijms-27-00199-f004:**
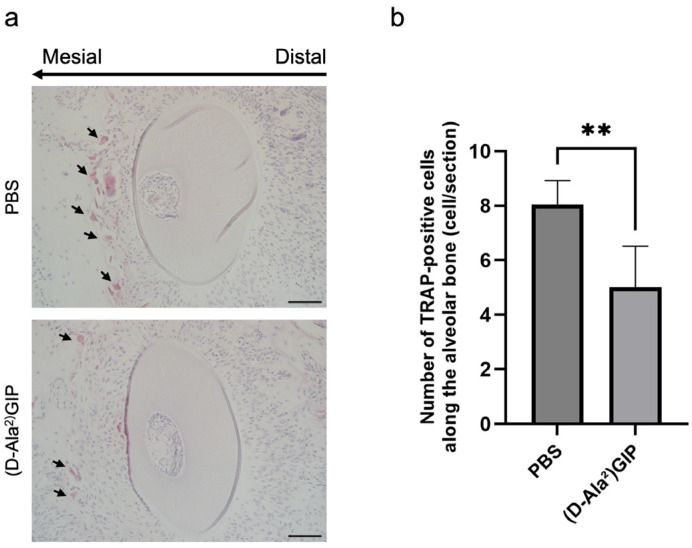
(D-Ala^2^)GIP inhibited osteoclast formation on the compression side during tooth movement. (**a**) Representative TRAP-stained histological sections of M1DB after 12 days of orthodontic tooth movement in PBS and (D-Ala^2^)GIP groups. Sections were counterstained with hematoxylin. The long arrow indicates the direction of tooth movement, and the small arrows point to osteoclasts on the compression side. Scale bar = 50 μm. (**b**) Quantification of TRAP-positive multinucleated cells along the alveolar bone surface surrounding M1DB following 12 days of orthodontic force application in PBS and (D-Ala^2^)GIP groups. Data are shown as mean ± SD. Statistical significance was determined using unpaired *t*-test (*n* = 4 mice/group; ** *p* < 0.01).

**Figure 5 ijms-27-00199-f005:**
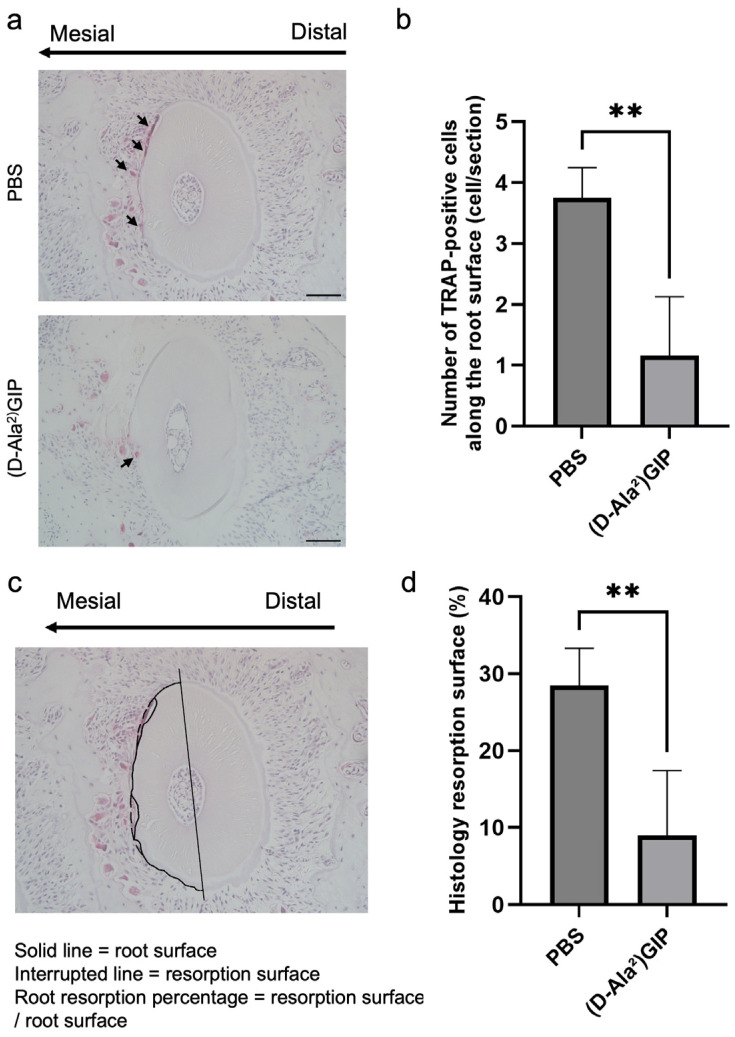
(D-Ala^2^)GIP inhibited odontoclast formation and root resorption during tooth movement. (**a**) Representative TRAP-stained histological sections of M1DB after 12 days of orthodontic tooth movement in PBS and (D-Ala^2^)GIP groups. Sections were counterstained with hematoxylin. The long arrow indicates the direction of tooth movement, and the small arrows point to osteoclasts on the compression side. Scale bar = 50 μm. (**b**) Quantification of TRAP-positive multinucleated cells along the surface of M1DB following 12 days of orthodontic force application in PBS and (D-Ala^2^)GIP groups. (**c**) A representative diagram showing the evaluation method of the root resorption surface percentage of M1DB. The solid line indicates the actual intact root surface, while the interrupted line indicates the resorption surface. The value of root resorption surface percentage was determined as the ratio of the resorbed surface to the root surface. (**d**) Ratio of root resorption surface of M1DB in histology sections after 12 days of orthodontic tooth movement in PBS and (D-Ala^2^)GIP groups. Data are shown as mean ± SD. Statistical significance was determined using unpaired *t*-test (*n* = 4 mice/group; ** *p* < 0.01).

## Data Availability

The original contributions presented in this study are included in the article. Further inquiries can be directed to the corresponding author.

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
