# Peer review of "(D-Ala2)GIP Inhibits TNF-α-Induced Osteoclast Formation and Bone Resorption, and Orthodontic Tooth Movement"

_ijms, 2025, doi:10.3390/ijms27010199_

Round 1
Reviewer 1 Report
Comments and Suggestions for Authors
The authors used the DPP-4-93 resistant GIP analog (D-Ala2)GIP to investigate the effects of exogenous and prolonged GIP signaling on TNF-α-induced osteoclastogenesis and bone resorption in vivo, as well as on orthodontic tooth movement and associated root resorption
The introduction is comprehensive, including studies conducted by the authors, with the aim of establishing the basis for the project.
The results are clearly expressed, easy to understand, and supplemented with graphs that facilitate comprehension of the study. However, the number of animals used in each result and the number of experiments conducted are not indicated.
The discussion is overly descriptive. The mechanisms that determine the effect of GIP in inhibiting osteoclastogenesis and increasing TNF-mediated resorption are not indicated. The comment made about tirzetapide is speculative and is not supported by the data presented. The strengths and weaknesses of the study should be included.
The methodology is described exhaustively and could be replicated by another group.
Author Response
To Reviewer 1:
Thank you very much for your constructive comments and suggestions. We sincerely appreciate them and will answer them in order.
- The authors used the DPP-4-93 resistant GIP analog (D-Ala2)GIP to investigate the effects of exogenous and prolonged GIP signaling on TNF-α-induced osteoclastogenesis and bone resorption in vivo, as well as on orthodontic tooth movement and associated root resorption
The introduction is comprehensive, including studies conducted by the authors, with the aim of establishing the basis for the project.
The results are clearly expressed, easy to understand, and supplemented with graphs that facilitate comprehension of the study. However, the number of animals used in each result and the number of experiments conducted are not indicated.
Thank you for pointing this out. We agree that the “n = 4” description in the figure legends was not sufficiently explicit regarding the number of animals and the experimental replicates. We have now revised all figure legends to state “n = 4 mice/group” and clarified this information in the Materials and Methods section (Section 4.2). We also added a statement indicating that each in vivo experiment was performed once, with individual animals treated as biological replicates.
- The discussion is overly descriptive. The mechanisms that determine the effect of GIP in inhibiting osteoclastogenesis and increasing TNF-mediated resorption are not indicated.
Thank you for this important comment. We agree that the underlying mechanisms by which GIP suppresses osteoclastogenesis and TNF-α–mediated resorption were not sufficiently discussed in the original version. We have now revised the Discussion section to clarify potential direct effects of GIP on osteoclast precursors, as well as possible indirect contributions from immune cells and osteoblast-lineage cells, based on prior in vitro findings. In addition, we now discuss the possibility that GIP may modulate both osteoclast formation and osteoclast activity, which together could contribute to the attenuation of TNF-α-mediated resorptive responses.
- The comment made about tirzetapide is speculative and is not supported by the data presented.
Thank you for pointing this out. We agree that our previous statements regarding tirzepatide were speculative and not directly supported by the data presented. In response, we have revised the Discussion and Conclusion sections to moderate the interpretation and clarify that the present study does not provide direct evidence for the effects of tirzepatide or other GIP agonists on orthodontic outcomes under systemic administration. Instead, our findings highlight the potential biological involvement of GIP signaling in bone responses associated with orthodontic force and inflammatory conditions, and provide a rationale for future studies examining the effects of systemic GIP-related therapies in clinical contexts.
- The strengths and weaknesses of the study should be included.
Thank you for this helpful suggestion. We have now added a summary in the Discussion section on the major strengths and limitations of the study. Specifically, we highlight that a key strength of this work is being the first to directly evaluate the effects of exogenous GIP administration on orthodontic tooth movement as well as on TNF-α–associated inflammatory bone resorption in vivo. We also clearly acknowledge several important limitations, including the lack of detailed mechanistic investigation into how GIP inhibits TNF-α–induced osteoclastogenesis and the use of a local injection model rather than systemic drug administration. These points are now discussed to better frame the scope of the current findings and to emphasize directions for future studies.
The methodology is described exhaustively and could be replicated by another group.

Reviewer 2 Report
Comments and Suggestions for Authors
The manuscript investigates the effects of (D-Ala²)GIP on TNF-α–induced osteoclastogenesis, bone resorption, and orthodontic tooth movement (OTM) in mice. The topic is timely and relevant, given the increasing clinical use of GIP-based therapeutics. The experiments are generally well designed, and the findings provide novel insights. However, several areas require clarification.
- The in vivo findings are clear, but the mechanistic interpretation remains speculative. The authors should add data or discussion addressing whether GIPR is expressed on osteoclast precursors in the periodontal ligament during OTM. Furthermore, the authors should discuss whether (D-Ala²)GIP may also act indirectly via osteoblasts/osteocytes or immune cells, not only osteoclast precursors.
- The manuscript suggests that tirzepatide or GIP agonists may affect orthodontic outcomes. The authors should emphasize limitations of extrapolating mouse gingival injection to systemic human drug exposure.
- The description of measuring resorption surface ratio needs more details.
- Some figures require major resolution (Fig.1,4,5).
- In the methods, the description “buccal gingiva adjacent to the molar” is vague. Provide a standardized anatomical description.
- Standardize gene/protein name formatting (e.g., italic for gene symbols) in all the manuscript.
The manuscript presents interesting and potentially important results but requires substantial clarification, methodological strengthening, and deeper mechanistic discussion. The revisions above are necessary before the work can be considered for publication.
Author Response
To Reviewer 2:
Thank you very much for your constructive comments and suggestions. We sincerely appreciate them and will answer them in order.
- The manuscript investigates the effects of (D-Ala²)GIP on TNF-α–induced osteoclastogenesis, bone resorption, and orthodontic tooth movement (OTM) in mice. The topic is timely and relevant, given the increasing clinical use of GIP-based therapeutics. The experiments are generally well designed, and the findings provide novel insights. However, several areas require clarification.
The in vivo findings are clear, but the mechanistic interpretation remains speculative. The authors should add data or discussion addressing whether GIPR is expressed on osteoclast precursors in the periodontal ligament during OTM. Furthermore, the authors should discuss whether (D-Ala²)GIP may also act indirectly via osteoblasts/osteocytes or immune cells, not only osteoclast precursors.
Thank you for this important comment. We agree that direct evidence regarding GIPR expression on osteoclast precursors during OTM would be valuable. However, direct identification of osteoclast precursors in the periodontal ligament during orthodontic force application is technically challenging, due to their transient nature and the absence of highly specific markers that reliably distinguish osteoclast precursors from other mononuclear cell populations. Therefore, direct assessment of GIPR expression on osteoclast precursors in this in vivo OTM model was difficult to achieve with sufficient specificity in the present study. Instead, we have expanded the Discussion to explicitly acknowledge the lack of direct evidence for GIPR expression on osteoclast precursors during OTM. We now discuss potential direct effects of GIP on osteoclast precursors, supported by our previous in vitro findings demonstrating that (D-Ala²)GIP suppresses TNF-α–induced osteoclastogenesis in osteoclast precursor monocultures. In addition, we outline possible indirect mechanisms mediated by immune cells and osteoblast-lineage cells, based on prior in vitro studies showing that (D-Ala²)GIP inhibits LPS-induced TNF-α expression in macrophages and reduces RANKL expression in osteoblasts. Finally, we clarify the limitations of extrapolating these in vitro observations to the current in vivo models and emphasize the need for future mechanistic investigations.
- The manuscript suggests that tirzepatide or GIP agonists may affect orthodontic outcomes. The authors should emphasize limitations of extrapolating mouse gingival injection to systemic human drug exposure.
Thank you very much for this thoughtful comment. We agree that careful consideration is required when interpreting the potential clinical relevance of our findings. Accordingly, we have revised the Discussion and Conclusion sections to tone down speculative statements regarding tirzepatide and to clearly emphasize the limitations of extrapolating results obtained from a local-site injection model in mice to systemic drug exposure in humans. We now explicitly state that the present findings do not provide direct evidence for the clinical effects of tirzepatide or other GIP agonists on orthodontic outcomes, but rather offer a biological framework for future investigation.
- The description of measuring resorption surface ratio needs more details.
Thank you for pointing this out. We have modified the manuscript and provided more details accordingly in the Materials and Methods section (Section 4.2). Specifically, we clarified that measurements were performed on transverse sections of the mesial (compression) surface of the distobuccal root of the maxillary first molar, with the total mesial root surface length traced along the cementum-periodontal ligament interface and resorption surface traced separately within the same contour using ImageJ (solid and interrupted lines in Fig. 4c). The resorption surface percentage was calculated as the ratio of resorption surface length to the total mesial root surface length.
- Some figures require major resolution (Fig.1,4,5).
Thank you for pointing this out. We have revised Figures 1, 4, and 5 by replacing them with higher-resolution images to improve visual clarity.
- In the methods, the description “buccal gingiva adjacent to the molar” is vague. Provide a standardized anatomical description.
Thank you for pointing this out. The reagents were injected into the gingivobuccal fold of the upper-left first molar, and we have revised the manuscript in Materials and Methods section (Section 4.5) to provide this clearer anatomical description. The corresponding description in the Discussion section has also been revised to ensure consistency throughout the manuscript.
- Standardize gene/protein name formatting (e.g., italic for gene symbols) in all the manuscript.
Thank you very much for this helpful suggestion. In this manuscript, TRAP and cathepsin K are referred to by their commonly used protein names, which are widely employed as osteoclast markers in the literature, rather than by their corresponding gene symbols (Acp5 and Ctsk). Accordingly, these terms are written in non-italic font. GAPDH is likewise presented in uppercase for consistency. The only other related term appearing in the text is GIPR, which refers to the receptor protein rather than the gene Gipr. Because no gene symbols are used in the manuscript text, italic formatting was not applicable; however, we have carefully checked the manuscript to ensure consistent nomenclature throughout and appreciate the reviewer’s comment.
The manuscript presents interesting and potentially important results but requires substantial clarification, methodological strengthening, and deeper mechanistic discussion. The revisions above are necessary before the work can be considered for publication.

Round 2
Reviewer 1 Report
Comments and Suggestions for Authors
The questions have answered by the authors
Reviewer 2 Report
Comments and Suggestions for Authors
The manuscript can be accepted after this revision